# Multiple Risks and Adolescent Depressive Symptoms in Ethnic Regions of China: Analyses Using Cumulative Risk Model, Logistic Regression, and Association Rule Mining

**DOI:** 10.3390/bs15050657

**Published:** 2025-05-12

**Authors:** Ting Zhou, Chen Wang, Jennifer Hu, Shan Zhang, Lin Fu, Zheng Huang, Huiying Qi

**Affiliations:** 1Department of Medical Psychology, School of Health Humanities, Peking University, Beijing 100191, China; zhouting.92@bjmu.edu.cn (T.Z.); jennifer_hu@stu.pku.edu.cn (J.H.); 2311220022@stu.pku.edu.cn (S.Z.); 2Department of Health Informatics and Management, School of Health Humanities, Peking University, Beijing 100191, China; wangchenparis@bjmu.edu.cn; 3School of Sociology, Beijing University of Technology, Beijing 100124, China; linfu@bjut.edu.cn; 4Department of Psychology, University of Chinese Academy of Sciences, Beijing 100101, China; 5Institute of Psychology, Chinese Academy of Sciences, Beijing 100101, China

**Keywords:** depressive symptoms, adolescents from ethnic minority areas, cumulative risk model, logistic regression, association rule mining

## Abstract

The present study aimed to examine the relationship between multiple risk exposures in family and school settings and the depressive symptoms of Chinese students in early adolescence living in the ethnic regions of Yunnan and Hebei, China, via different multiple risk analytic approaches. A total of 2940 students (47.3% females) in grades 4 to 9 (*M*age = 12.08, *SD* = 2.04) from ethnic minority counties in Yunnan and Hebei participated in the survey. The participants completed the questionnaires and reported family risk, school risk, depressive symptoms, and demographic information. The cumulative risk model and the individual multiple risk models with logistic regression/association rule mining were used to examine the effects of cumulative risk, the relative contributions of individual risks, and combinations of multiple risks. We found that (1) when a cumulative risk model was used, the associations between family cumulative risk and school cumulative risk on depressive symptoms were significant, but the cross-domain interaction effect was not significant. (2) The results of logistic regression indicated that high levels of family conflict, low levels of family cohesion, low levels of teacher support, and low levels of peer support were significantly correlated with a high risk for depression. (3) The results of association rule mining revealed meaningful associations between multiple risk factor combinations and depressive symptoms. In conclusion, the use of association rule mining enhanced the analyses and understanding of the effects of multiple risk exposures. Interpersonal stressors in family and school settings need to be addressed in depression prevention and intervention programs for adolescents.

## 1. Introduction

Adolescents living in multiethnic areas are exposed to more culturally diverse environments and usually face specific stressors related to acculturation that might influence their mental health. However, the relationship between environmental factors and depressive symptoms is less well understood in this population.

[7] ([7]) proposed a cultural–ecological framework to understand the environmental risk factors for developmental competences in minority children. This theory underscores the importance of examining multiple risk factors in various domains. We focused on multiple risk factors in family and school settings for two reasons. First, from the ecological perspective of children’s development ([6]), family and school environments are the most salient environments for adolescents. Second, school is an appropriate setting to deliver mental health prevention and early intervention programs for adolescents. Parental education programs could also be delivered with the assistance of the school. Thus, identified risk factors could be targeted in school-based or family-based intervention programs to improve the mental health of adolescents.

Family risk factors for adolescent mental health have been well studied in previous research. Low family socioeconomic status, single-parent status, and separation from parents are typical familial risk factors ([11]). In addition, family cohesion and conflict are the most common indicators of family relationships and are associated with adolescent depressive symptoms ([31]; [41]). School is another crucial social environment for adolescents in which they spend most of the day. School is also an important socializing agenda where multiethnic students learn the customs, cultures, and lifestyles of different ethnicities and develop multiculturalism values. School support systems are considered especially important for ethnic minority adolescents. Teacher support is associated with reduced depressive symptoms for adolescents as a whole ([13]; [22]) and for ethnic minority adolescents specifically ([3]). Peer support also plays an important role in establishing self-identity and has protective effects on adolescents’ mental health ([19]; [40]). Providing opportunities for students to make their own choices promotes internal motivational resources ([16]; [29]). A lack of support from teachers and peers and low autonomy support are considered risk factors for depressive symptoms in the school setting. Regarding the role of ethnicity in predicting mental health outcomes, previous findings have been inconsistent. Some studies have found that ethnic minority status is associated with a higher risk of mental health problems ([18]; [36]), while others have reported no significant relationship between ethnicity and mental health outcomes ([42]). Therefore, the present study aims to clarify this issue by examining the effects of ethnicity in combination with family and school risk factors.

Environmental risk factors usually covary ([10]). Examining the effects of multiple risk factors in various domains could improve the variance of the criterion variable explained by the analytic model ([11]; [28]). The cumulative risk model and the multiple individual risks model are usually used to examine the effects of multiple risk factors. For the cumulative risk model, an aggregate metric of risk factor exposure is created by generating a composite metric wherein a set of dichotomous risk factor exposures are summed together ([11]). There is evidence that family cumulative risk (CR) and school CR are significantly associated with adverse developmental outcomes ([9]; [24]; [38]), but most studies have explored the effects of CR in one domain, leaving cross-domain risks less examined ([37]).

Although the cumulative risk model is commonly used, it assumes that the predictive effects of singular risk are identical, which is not true in most cases. The multiple individual risk factors model overcomes this drawback by recognizing the relative contributions of each risk factor ([43]). The traditional approach to building a multiple individual risk factors model is to perform multiple regressions or logistic regressions. The inclusion of independent variables requires a theoretical basis, which might lead to the omission of variables that are not explained by existing theories. Association rule analysis, a machine learning method, relies on data mining methods to analyze interrelationships in large-scale datasets, making it easier and more effective to obtain correlations between multiple variables and thereby identifying the patterns in the cooccurrences of risk factors.

The present study used both the cumulative risk model and the multiple individual risk factors model, utilizing logistic regression/association rule mining simultaneously, to obtain a more comprehensive understanding. The present study extends the literature by (1) considering the association of cross-domain risk in family and school settings and adolescent depressive symptoms; (2) examining the effects of multiple risk factors via different modeling techniques to overcome the limitations of individual techniques; and (3) focusing on the effects of multiple risk exposures on the depressive symptoms of adolescents living in the ethnic regions of China.

## 2. Methods

### 2.1. Participants and Procedure

This study was approved by our institute’s Ethics Committee. The required sample size was estimated a priori via PASS 15.0. For the cumulative risk model (multiple linear regression with 8 predictors), a minimum of 85 participants was needed to detect an effect size (f^2^ = 0.15) with 80% power (1 − β = 0.80) and a two-sided Type I error rate (α = 0.05).

The sample size required for the multiple logistic regression was also calculated. The examination of a univariate model required 298 participants to detect an odds ratio (OR) of 1.5 when the expected baseline probability (P0) was set as 0.2 (α = 0.05, power = 0.80). To account for the nine risk factors of interest, five control variables, and their potential collinearity, we incorporated a variance inflation factor (VIF) based on an assumed average pairwise correlation (R^2^ = 0.2) among predictors ([14]). The adjusted sample size was calculated as follows:N_adjusted_ = N × (1 + k × R^2^) = 298 × (1 + 14 × 0.2) = 1132,
where k represents the number of covariates.

To ensure robustness, sensitivity analyses were performed for varying R^2^ values (0.1–0.3), yielding a range of 715–1550 participants. Considering potential nonresponse and missing data, we increased the target sample size to N = 2000.

Students were recruited from 38 primary schools and 11 secondary schools in two ethnic minority counties, Yunnan and Hebei, via cluster sampling. Informed consent was obtained from the adolescents and their parents. Students completed paper-based questionnaires in class under the supervision of trained teachers. A total of 3200 adolescents were recruited, with 2940 completing the questionnaires. The response rate was 91.9%. This sample size met the requirements for statistical analyses. Data were entered into Epidata 2.1, with double entry conducted.

In the final sample, the adolescents’ ages ranged from 8 to 17 years (*M*age = 12.08, *SD* = 2.04). Among them, 495 (16.8%) students were in grade 4, 590 (20.1%) were in grade 5, 574 (19.5%) were in grade 6, 414 (14.1%) were in grade 7, 418 (14.2%) were in grade 8, and 449 (15.3%) were in grade 9. There were 1391 (47.3%) girls and 1549 (52.7%) boys. As a multiethnic sample, the majority of participants (n = 2498, 85%) were ethnic minorities (16 ethnicities, including Man, Bai, Lisu, Yi, Pumi, Hui, Miao, Nu, etc.), and 443 of them were Han Chinese (15.1%). Most participants had siblings (n = 2568, 87.3%), and 372 (12.7%) were the only child in the family.

### 2.2. Measures

**Demographic characteristics:** Demographic information such as gender, grade, ethnicity, and number of children in the family was collected.

**Adolescent depressive symptoms:** The Chinese version of the Depression Self-Rating Scale for Children (DSRSC) was used ([5]). This 18-item scale assesses depressive symptoms in children aged 8 to 16 years. The items are rated on a 3-point scale (0 = never, 1 = sometimes, 2 = often). The DSRSC has demonstrated satisfactory reliability and validity. A cutoff point of 15 has been suggested to distinguish adolescents at low and high risk of depression ([30]). The internal consistency was acceptable (α = 0.82).

**Family risk factors:** Six family risk factors were measured. Family structure was measured by the item “With whom do you live?”. Separation from parents was measured by the item “I separated from my parents for the past six months”. The educational level of the father and mother was measured via one item. Family financial hardship was measured via the Chinese version of the family economic strain scale ([34]), which measures the frequency of specific economic problems faced in the past year. The items were rated on a 5-point Likert scale (1 = never and 5 = always), and an example item is “My family does not have enough money to buy the food I like” (α = 0.79). Family cohesion was measured via the 16-item cohesion subscale of the Family Adaptation and Cohesion Evaluation Scales II-Family version (FACES-II) ([25]). An example item is “Family members feel very close to each other”. The items are rated on a 5-point Likert scale (1 = almost never; 5 = almost always). The Chinese version of the FACES-II has good reliability and validity ([12]), and the Cronbach’s α was 0.83. Family conflict was measured via the 9-item conflict subscale of the Family Environment Scale ([23]). An example item is “Family members often blame and criticize each other”. The participants rated the items on a dichotomous scale (no = 0; yes = 1). The Chinese version of this scale has been validated ([12]; [32]). Similarly to the results of the study examining the psychometric properties of the Chinese version of the FES (for the conflict subscale, Cronbach’s α = 0.64; Tao et al., 2015), the present study yielded a Cronbach’s α of 0.61. The criteria for risk are displayed in Table 1.

**School risk factors:** Low levels of teacher support, peer support and autonomy support were used to indicate school risk factors. All these factors were measured via the school climate scale ([15]). There were 7 items used to measure teacher support (e.g., “teachers believe I can do well”), 13 items used to measure peer support (e.g., “students care about one another”) and 5 items used to measure autonomy support (e.g., “students are given the chance to help make decisions”). The items are rated on a 4-point Likert scale (1 = never; 4 = always). The Cronbach’s α values were 0.81, 0.78, and 0.75 for each subscale. The criteria for risk are displayed in Table 1.

#### Analytic Approaches

***The cumulative risk model:*** Family CR was calculated by summing the risk scores of the six family risk factors, and school CR was calculated by summing the risk scores of the three school risk factors. The level of adolescent depressive symptoms was used as a continuous variable in this model. Hierarchical regression was performed to examine the main effects of family CR, school CR, and their interaction effects on adolescent depressive symptoms.

***The multiple individual risk factors model using logistic regression analysis***: Univariate logistic analyses were first conducted to screen for factors significantly related to adolescent depressive symptoms, which were treated as dichotomous variables on the basis of the cutoff for depression risk status in the DSRSC. The significant risk factors were included in the multiple logistic regression to examine the effects of multiple individual risk factors. For both the cumulative risk model (multiple regression) and the multiple individual risk factors model (logistic regression), multicollinearity was evaluated via tolerance (critical value: <0.2) and the variance inflation factor (VIF, critical value: >5) ([21]).

***The multiple individual risk factors model using association rule mining***: The form of the association rule is X ⇒ Y, which represents the association in an “if–then” format. X denotes a subset of all predictor variables, and Y denotes the outcome variables, which are referred to as the left-hand side (LHS) and right-hand side (RHS) of the association rule ([1]). All the variables were dichotomous in this model.

The association rule evaluates the association strength through support, confidence, and lift. The definitions and formulas of these evaluation parameters are shown in Table 2.

The association rule X⇒Y is considered valuable when both the support (sup(X⇒Y)) and confidence (conf(X⇒Y)) exceed or equal the predefined threshold values. These threshold values, namely, min_sup and min_conf, represent the minimum significance and reliability requirements for the association rules.

The Apriori algorithm of association rules was employed in this study to identify risk factors associated with depression through an analysis of the relationships between predictive variables and outcomes ([2]). The implementation process consists of two parts: (1) discovering frequent item sets, defined as item sets with a support greater than or equal to the given min_sup, and (2) generating association rules. Subsequently, effective strong association rules were obtained on the basis of different support, confidence, and lift parameters, followed by an analysis of the influencing factors behind the relevant results.

## 3. Results

### 3.1. Descriptive Analysis

As displayed in Table 3, adolescents’ depressive symptoms were significantly and positively correlated with financial strain (*r* = 0.14, *p* < 0.001) and family conflict (*r* = 0.40, *p* < 0.001). The level of depressive symptoms showed significant negative correlations with family cohesion, teacher support, peer support, and autonomy support, with *r*s values ranging from −0.27 to −0.48, *p*s < 0.001.

### 3.2. Approach 1: Cumulative Risk Model

The results of the descriptive analyses of individual risk factors are shown in Table 1. The average score of family CR was 1.29 (SD = 1.28), with scores ranging from 0 to 6. The average school CR score was 0.64 (SD = 0.88), with scores ranging from 0 to 3. The assumptions of multiple linear regression were tested and met. The residuals of the dependent variable (depressive symptoms) were approximately normally distributed, as indicated by skewness (0.63) and kurtosis (0.71) values both less than 1. Scatter plots showed that the relationships between the continuous independent variables and the dependent variable were approximately linear. Homoscedasticity was supported by the residuals versus predicted values plot, which showed an even dispersion of points without any clear pattern, indicating that the variance of the residuals was roughly equal. The multicollinearity of independent variables was examined by calculating the tolerance and VIF. The range of tolerance was from 0.90 to 1.00, and the range of VIF was from 1.00 to 1.11, indicating negligible multicollinearity problems. The hierarchical regression indicated that this model explained 21.1% of the variance in adolescent depressive symptoms. Family CR and school CR were positively associated with adolescent depressive symptoms, with *β* = 0.22 and *p* < 0.001, and β = 0.33 and *p* < 0.001, respectively, but the predictive effect of the interaction term was not significant, *F*(1, 2931) =0.03, Δ*R*^2^ < 0.001, and *p* = 0.859.

### 3.3. Approach 2: Multiple Individual Risk Factors Model via Logistic Regression Analysis

The results of univariate logistic regressions indicated that gender, grade, and academic rank were significantly associated with the risk of depression. Specifically, girls were at a greater risk of depression than boys were (OR = 1.40, *p* < 0.001). Students in higher grades were at a greater risk of depression than students in lower grades (OR = 1.09, *p* < 0.001). Students with poorer academic performance had a greater risk of depression (OR = 1.15, *p* < 0.001). With respect to family risk factors, not living with one’s parents, separation from one’s parents, suffering from family financial strain, experiencing low family cohesion, and experiencing high levels of family conflict were significant risk factors for adolescent depression. With respect to school risk factors, low teacher support, peer support, and autonomy support were all significant risk factors for adolescent depression. The regression coefficients are displayed in detail in Table 4.

Five family risk factors (not living with parents, separation from parents, family financial strain, low family cohesion, and high family conflict), three school risk factors (low teacher support, low peer support, and low autonomy support) and three control factors (gender, grade, and academic ranking) were included in the multiple logistic regression. The potential multicollinearity among predictors was evaluated via the linear regression approximation method. The tolerance and variance inflation factor (VIF) were computed. Following the established thresholds (tolerance > 0.2 and VIF < 5), no significant multicollinearity was detected (the tolerance ranged from 0.50 to 0.96, and the variance inflation factor (VIF) ranged from 1.04 to 1.69).

The results indicated that this model explained 24.2% of the variance in adolescents’ depression risk status. The results revealed several significant associations with adolescent depression risk status. Specifically, gender was a significant predictor, with girls at a greater risk of depression (*B* = 0.54, *SE* = 0.10, *p* < 0.001, OR = 1.74, 95% *CI* [1.44, 2.07]). Additionally, being in a higher grade was associated with an increased risk of depression (*B* = 0.07, *SE* = 0.03, *p* = 0.033, OR = 1.07, 95% *CI* [1.01, 1.12]). Family cohesion was negatively associated with depression risk status, suggesting that adolescents from families with stronger bonds were less likely to experience depressive symptoms (*B* = −0.61, *SE* = 0.01, *p* < 0.001, OR = 0.94, 95% *CI* [0.93, 0.95]). Conversely, high family conflict was positively associated with an increased risk for depression, indicating that adolescents in conflict-ridden families were at a greater risk for depression (*B* = 0.23, *SE* = 0.03, *p* < 0.001, OR = 1.26, 95% *CI* [1.20, 1.33]). In terms of school-related factors, both teacher support and peer support were protective factors against adolescent depression. Specifically, higher levels of teacher support were associated with a lower risk of adolescent depression (*B* = −0.06, *SE* = 0.01, *p* < 0.001, OR = 0.94, 95% *CI* [0.92, 0.97]), as were higher levels of peer support (*B* = −0.11, *SE* = 0.01, *p* < 0.001, OR = 0.90, 95% *CI* [0.88, 0.91]).

### 3.4. Approach 3: Multiple Individual Risk Factors Model Using Association Rule Mining

To identify the risk factors associated with depression risk status, RHS was set as depression to establish association rules. By taking lift (X⇒Y) > 1 as the basic requirement, we tested various combinations of min_sup and min_conf. We observed that when min_sup > 5% or min_conf > 65%, the number of association rules obtained was less than 10, which may lead to the omission of some valuable rules. Conversely, when min_sup < 5% or min_conf < 65%, the number of association rules increased significantly, containing many redundant rules, and the significance and reliability of the association rules decreased. On the basis of a comprehensive evaluation of the quantity and quality of the rules, we finally applied min_sup = 5% and min_conf = 65%. The generated association rules are arranged in descending order of the confidence degree, as shown in Table 5. Support indicates the frequency of the co-occurrence of risk factors and depression, and confidence indicates the frequency of depression occurring under the conditions of risk factors. Lift indicates the relationship between risk factors and depression, with a value greater than 1 indicating a positive relationship between the two.

A total of 18 strong association rules were derived, including two 5-item sets, nine 4-item sets, and seven 3-item sets. The support for the depression association ranged from 5.21% to 7.52%, and the confidence ranged from 65.37% to 82.14%. The four risk factors among the nine risk factors used for LHS that are included in the association rules are high family conflict, low family cohesion, low peer support, and low teacher support. Additionally, there was a strong association between demographic characteristics (such as being from multiple-child families, ethnic minorities, females, and secondary school) and depression. The combination of high family conflict, low family cohesion, and being from a multiple-child family had the strongest correlation with depression (lift = 2.78).

## 4. Discussion

Given that familial and school-related factors are closely correlated with adolescent depressive symptoms and considering the concurrent presence of multiple risk factors in real-world settings, it is important to evaluate the association of exposure to multiple risks and adolescents’ mental health. The present study addressed this issue by examining the relationship between risk exposure in family and school settings and the depressive symptoms of adolescents living in ethnic minority areas in China. The cumulative risk model and the multiple individual risk factors model with logistic regression/association rule mining were utilized to examine the overall effects of multiple risk exposures, clarify the relative contribution of each risk factor, and explore the effects of risk factor combinations.

Generally, different analysis approaches yield some common results. First, we confirmed that both family risk factors and school risk factors uniquely correlated with depressive symptoms in adolescents living in a multiethnic environment. According to the cumulative risk model (Approach 1), both family CR and school-level CR are significantly related to adolescent depressive symptoms. Using the multiple individual risks model (Approaches 2 and 3), significant risk factors were found in both domains. This result was in line with Bronfenbrenner’s theory, which addresses the importance of family and school microsystems in children’s psychosocial development and supports the cumulative risk theory that multiple risks better explain development outcomes than singular risk factors do. Therefore, it is important to identify multiple risk factor combination patterns that predict depression. Second, risk factors identified via logistic regression (Approach 2) and association rule mining (Approach 3) showed similarities. Low family cohesion, high family conflict, low teacher support, and low peer support were significant risk factors found via multiple logistic regression (Approach 2), and they were also strongly positively associated with depression risk status according to association rule mining (Approach 3). The results underscore the importance of interpersonal stressors in family and school settings in predicting adolescent depression risk status.

A cohesive family climate is associated with higher levels of perceived social support ([35]). In contrast, family conflict is related to increased adolescent stress ([17]), greater emotional insecurity ([4]), and potential difficulties in the development of emotion regulation abilities ([8]), which may, in turn, be linked to poorer mental health outcomes ([4]; [20]). This result was similar to the findings of [39]’s ([39]) meta-analysis, in which parental warmth and interparental conflict were found to be the most well-documented predictors of youth depressive and anxious symptoms among various parental factors ([39]). Therefore, family cohesion and conflict may be important factors to consider in family-based programs aiming to prevent or reduce depressive symptoms for adolescents.

A lack of teacher support and peer support was found to be strongly associated with adolescent depressive symptoms. The results expand the findings of previous research focusing on the effects of teacher support or peer support only ([19]; [22]; [39]; [40]) by indicating the pronounced effects of teacher support and peer support among multiple factors and highlighting the potential value of improving school connectedness (both relationships with teachers and peers) in efforts to reduce depression risk among adolescents. It is notable that the observed correlations between interpersonal factors in family and school settings and adolescent depressive symptoms could also be interpreted in the reverse direction. Depressed adolescents might receive less social support because of impaired social functioning, and they might also perceive less social support because of negative bias in social cognition ([26]). The causality could not be examined using the current cross-sectional design and should be further examined using a longitudinal design in the future.

Using association rule mining, we found that a combination of multiple factors had stronger correlations with depression risk status, which were specific findings from Approach 3. Some factor combinations were within one domain; for example, the combination of high family conflict and low family cohesion had a positive correlation with depression risk. There are also significant cross-domain combinations; for example, the combination of low family cohesion and low peer support was positively correlated with adolescent depression risk. Although gender and grade themselves were not identified as separate association rules using the predefined threshold, their roles cannot be ignored. Female gender and higher grade increased the effects of other risk factors. As indicated by the results of Approach 3, the combination of female gender and low peer support/high family conflict was correlated with depression risk. That is, the association between high levels of family conflict/low levels of family cohesion and depression risk appeared to be stronger among girls. This result is in line with previous findings on gender differences in vulnerability to depression ([27]). Regarding the role of age/grade, we found that the combination of secondary school attendance and low family cohesion was correlated with adolescent depression, whereas low family cohesion itself was not correlated with depression. This result is consistent with the fact that the incidence of depression in adolescence is greater than that in childhood. The greater susceptibility of secondary school students might be related to pubertal development and increasing social demands ([33]).

Regarding the role of ethnicity, although its association with depressive symptoms was not significant when Approaches 1 and 2 were used, ethnic minorities appeared in several association rules to predict depression risk when Approach 3 was used. Specifically, we found that when combined with ethnic minorities, the correlations with depression increased for the factors “low peer support, low family cohesion, from multiple-child families”, “low peer support and low family cohesion”, “secondary school and low family cohesion”, “low peer support, female, from multiple-child families”, and “low peer support, female”, whereas the correlation with depression slightly decreased when ethnic minorities were combined with high family conflict and low family cohesion. Generally, the results suggested that ethnic minority students may be more likely to experience depressive symptoms when exposed to multiple familial and school-related factors; however, being an ethnic minority alone is not directly associated with depression risk status. This finding aligns with those of several studies ([42]), but it contrasts with other studies suggesting that ethnic minority status can be an independent risk factor for children’s development ([18]; [36]). From the perspective of the cultural–ecological framework, the effects of ethnicity depend on specific cultural and ecological environments. People living at the sampling sites have a long history of multiethnic integration. Folk culture and life customs are well preserved and respected. Children learn Mandarin from an early age and face no language barriers in cross-ethnic communications. These factors might buffer the negative effects of ethnicity-related stressors to some extent. However, because the majority of participants were ethnic minority students, the class imbalance of the dataset might have led to bias in the results. In addition, because we did not measure ethnicity-related stressors directly, this issue should be explained with caution and needs further examination.

### 4.1. Implications

This study contributes to research methods on multiple risk exposures and mental health. By comparing the results of multiple analyses, we found that association rule mining has advantages in identifying significant effects from a number of factor combinations.

Our results also have practical implications for educators in ethnic regions in terms of mental health screening and intervention development. First, individuals with higher grades, girls, those exposed to family risk factors such as family conflict and low cohesion, and those exposed to school risk factors such as low teacher support tended to report more severe depressive symptoms. A combination of multiple risk factors was associated with higher levels of depressive symptoms. Second, this study provides direct evidence on the targets and forms of depression prevention and intervention programs for adolescents living in ethnic regions.

### 4.2. Limitations and Future Directions

This study has some limitations. First, only a few risk factors in the family and school domains were involved, while other risk factors, such as teacher experience and school bullying, were overlooked. In addition, the effects of individual factors such as personality and physical health were not considered. Future research might involve more elements in Bronfenbrenner’s ecological theory for a more comprehensive understanding. Second, ethnicity-related risk factors were not directly measured, which should be addressed in further studies on children’s mental health in ethnic regions. Third, this study’s cross-sectional design limited its ability to infer causality. Therefore, this issue should be examined in the future via a longitudinal design.

## 5. Conclusions

Family and school risk factors were significantly associated with an increased risk of depression among students in early adolescence living in ethnic minority areas in China. Combinations of risk factors were linked to a higher likelihood of depression and predicted depression risk in an interactive manner. Interpersonal stressors, including low family cohesion, high family conflict, and low teacher support and peer support, were the most salient factors associated with adolescent depressive symptoms and should be considered in prevention and intervention programs.

## Figures and Tables

**Table 1 behavsci-15-00657-t001:** Risk factors, risk criteria, and percentage of participants at risk.

Risk Factor	Range of the Raw Score	Risk Criterion	At Risk (%)
*Family risk factors*			
Unfavorable family structure	0/1	Not living with parents	24.0%
Separation from parents	0/1	Have not lived with parents for 6 months because parents have been out at work.	36.8%
Low parental education	1–6	Parental education level was below high school (completing only middle school or less).	53.6%
Financial strain	1–20	The score of the family economic strain scale is above the 75th percentile.	26.7%
Low family cohesion	17–75	The score of the family cohesion subscale in FACES-II is below the 25th percentile.	14.7%
High family conflict	0–9	The score of the family conflict subscale in the FES is above the 75th percentile.	22.8%
*School risk factors*			
Low teacher support	7–28	The score of the teacher support subscale in the school climate scale is below the 25th percentile.	15.6%
Low peer support	3–20	The score of the peer support subscale in the school climate scale is below the 25th percentile.	24.6%
Low autonomy support	16–52	The score of the autonomy support subscale in the school climate scale is below the 25th percentile.	23.9%

Note: FACES-II = Family Adaptation and Cohesion Evaluation Scales II-Family version; FES = Family Environment Scale.

**Table 2 behavsci-15-00657-t002:** Definitions of the formula and explanations of support, confidence, and lift.

Parameter	Formula (P Is the Probability of an Association)	Meaning
Support	$\sup\left(X\Rightarrow Y\right)=P\left(X\cap Y\right)$	Support refers to the probability of {X, Y} appearing in all item sets; that is, the probability that the item set contains both X and Y.
Confidence	$\mathrm{conf}\left(X\Rightarrow Y\right)=P\left(Y|X\right)$	The confidence denotes the conditional probability of the occurrence of RHS Y given that LHS X occurs, i.e., the probability of including Y in the item set containing X.
Lift	$\mathrm{lift}\left(X\Rightarrow Y\right)=P\left(X\cap Y\right)/P\left(X\right)P\left(Y\right)$	The lift measures the dependency between X and Y, with values greater than 1 indicating a positive correlation and values less than 1 indicating a negative correlation.

**Table 3 behavsci-15-00657-t003:** Descriptive statistics and bivariate correlations of the main variables (N = 2940).

Variable	M (SD)	1	2	3	4	5	6	7
1—Financial strain	7.51 (3.72)	-						
2—Family cohesion	53.27 (9.56)	−0.17 ***	-					
3—Family conflict	2.47 (1.94)	0.12 ***	−0.41 ***	-				
4—Teacher support	21.85 (4.20)	−0.05 **	0.38 ***	−0.19 ***	-			
5—Autonomy support	14.24 (3.69)	−0.07 ***	0.33 ***	−0.12 ***	0.57 ***	-		
6—Peer support	39.58 (5.91)	−0.19 ***	0.39 ***	−0.27 ***	0.44 ***	0.43 ***	-	
7—Depressive symptoms	11.87 (5.94)	0.14 ***	−0.48 ***	0.40 ***	−0.36 ***	−0.27 ***	−0.45 ***	-

Note: ** *p* < 0.01, and *** *p* < 0.001.

**Table 4 behavsci-15-00657-t004:** The OR values of risk variables were determined via univariate logistic regression and multiple logistic regression.

Independent Variables	Univariate Logistic Regression			Multiple Logistic Regression		
	OR	95% CI	*p*	OR	95% CI	*p*
Gender	1.40	[1.20, 1.65]	<0.001	1.71	[1.44, 2.07]	<0.001
Grade	1.09	[1.04, 1.14]	<0.001	1.06	[1.01, 1.12]	0.033
Ethnicity	1.04	[0.84, 1.30]	0.704	-	-	-
Only child or not	1.07	[0.84, 1.37]	0.567	-	-	-
Academic rank	1.15	[1.07, 1.25]	0.000	1.03	[0.97, 1.17]	0.599
Not living with parents	1.22	[1.01, 1.46]	0.036	.89	[0.71, 1.11]	0.308
Separation from parents	1.31	[1.12, 1.55]	0.001	1.02	[0.83, 1.24]	0.884
Parental education risk	0.92	[0.79, 1.08]	0.328	-	-	-
Family financial strain	1.05	[1.03, 1.07]	<0.001	1.03	[0.84, 1.27]	0.791
Family cohesion	0.91	[0.90, 0.92]	<0.001	0.94	[0.93, 0.95]	<0.001
Family conflict	1.49	[1.42, 1.56]	<0.001	1.26	[1.20, 1.33]	<0.001
Teacher support	0.86	[0.84, 0.88]	<0.001	0.94	[0.92, 0.97]	<0.001
Peer support	0.86	[0.85, 0.87]	<0.001	0.90	[0.88, 0.91]	<0.001
Autonomy support	0.88	[0.86, 0.90]	<0.001	1.03	[0.99, 1.06]	0.152

**Table 5 behavsci-15-00657-t005:** Correlation analysis results for 5% min_sup and 65% min_conf.

LHS (RHS Was Depression)	Support	Confidence	Lift
high family conflict, low family cohesion, and from multiple-child family	5.48%	82.14%	2.78
high family conflict and low family cohesion	6.26%	81.06%	2.75
high family conflict, ethnic minority, and low family cohesion	5.21%	80.95%	2.74
low peer support, ethnic minority, low family cohesion, and from multiple-child family	5.61%	76.74%	2.60
low peer support, low family cohesion, and from multiple-child family	6.77%	76.25%	2.58
low peer support, ethnic minority, and low family cohesion	6.33%	75.00%	2.54
low peer support and low family cohesion	7.52%	74.41%	2.52
ethnic minority, secondary school, and low family cohesion	5.55%	71.18%	2.41
secondary school, low family cohesion, and from multiple-child family	6.09%	70.75%	2.40
low family cohesion and low teacher support	5.27%	70.14%	2.38
secondary school and low family cohesion	6.74%	69.96%	2.37
low peer support, ethnic minority, female, and from multiple-child family	5.55%	69.96%	2.37
low peer support, female, and from multiple-child family	6.67%	68.77%	2.33
low peer support, ethnic minority, and female	6.12%	68.70%	2.33
low peer support and low teacher support	5.48%	68.22%	2.31
low peer support and female	7.32%	67.82%	2.30
female, low family cohesion, and from multiple-child family	6.94%	66.45%	2.25
high family conflict and female	5.72%	65.37%	2.22

## Data Availability

The raw data supporting the conclusions of this article will be made available by the authors on request.

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
