# Peer review of "Multiple Risks and Adolescent Depressive Symptoms in Ethnic Regions of China: Analyses Using Cumulative Risk Model, Logistic Regression, and Association Rule Mining"

_behavsci, 2025, doi:10.3390/bs15050657_

Round 1
Reviewer 1 Report
Comments and Suggestions for Authors
Dear authors,
this is a very interesting and well drafted paper. Nevertheless, I do have some remarks and suggestions:
I do think in and for some of the tables more explanation could be useful:
Table 1: education of parents, could you indicate the cut-off for "below high school"
Table 3: you describe the negative correlations as "lowe to moderate" but they are all "stonger" than financial restrain; and family cohesion and peer support having the strongest (negative) correlastion
Table 4 seems not relevant at all (the only reference to table 4 is "the descriptive analyses of individual risk factors are shown in table one and 4" which in the case of table 4 looks wrong to me, since it shows only the distribution but not which CRs are combined.) I do not see the added value of table 4.
For table 5 the description of the multiple logistic regression is more or less putting the numbers of the table line by line in sentences - the description of the univarite version is better (e.g. students with poorer academic performance had a greater risk...) this kind of explanation is missing for the multiple logistic regression (most of the readers will understand the table alone, but the given description does not help those, who do not understand the table).
two more thing: please introduce the abbrevation CR. And can you explain why you have applied minsup=5% and min_conf=65%?
Regarding the discussion:
The presentation of the paper is focussing largely on a comparison of methods. I think this should get even more prominence in the discussion.
I think, it would be really interesting to discuss ethnicity in more detail (e,g, that it is not included in the multiple logistic regression) because the title of the paper somehow refers to ethnicity (you mention it only briefly in the limitations paragraph) but do the results for youth in ethnic minority regions differ from other regions in China?
Last point: references 4 and 5 are identical, the same holds for 11 and 12
Author Response
1. Table 1: education of parents, could you indicate the cut-off for "below high school"
ANSWER: We would like to clarify that parents who completed only a middle school education or less were identified as below high school. We also added the clarification in Table 1.
2. Table 3: you describe the negative correlations as "lowe to moderate" but they are all "stonger" than financial restrain; and family cohesion and peer support having the strongest (negative) correlastion
ANSWER: Thank you for this comment. In this paragraph, we reported positive correlations first and then reported negative correlations. A correlation coefficient is usually considered “low” when it is smaller than 0.3, and “moderate” when it is between 0.3-0.5. That is why we reported the negative correlations were low to moderate. We decided to remove “low to moderate” in the revised version to avoid confusion.
3. Table 4 seems not relevant at all (the only reference to table 4 is "the descriptive analyses of individual risk factors are shown in table one and 4" which in the case of table 4 looks wrong to me, since it shows only the distribution but not which CRs are combined.) I do not see the added value of table 4.
ANSWER: Thank you for this comment. We agreed that the information in Table 4 is limited; therefore, we removed this table in the revised manuscript. At the same time, we added descriptive analyses of family CR and school CR in the section of Results.
4. For table 5 the description of the multiple logistic regression is more or less putting the numbers of the table line by line in sentences - the description of the univarite version is better (e.g. students with poorer academic performance had a greater risk...) this kind of explanation is missing for the multiple logistic regression (most of the readers will understand the table alone, but the given description does not help those, who do not understand the table).
ANSWER: Thank you for this comment. We elaborated on results of the multiple logistic regression by providing more detailed interpretation on the results.
5. two more thing: please introduce the abbrevation CR. And can you explain why you have applied minsup=5% and min_conf=65%?
ANSWER: Thank you for this comment. Apologies for missing the explanation of CR. It was short for cumulative risk. The explanation was added.
In ARM, by taking lift (X⇒Y) > 1 as the basic requirement, we tested various combinations of min_sup and min_conf. We observed that when min_sup > 5% or min_conf > 65%, the number of association rules obtained was less than 10, which may lead to the omission of some valuable rules. Conversely, when min_sup < 5% or min_conf < 65%, the number of association rules increased significantly, containing a large number of redundant rules, and the significance and reliability of the association rules were diminished. Based on a comprehensive evaluation of the quantity and quality of the rules, we finally applied min_sup = 5% and min_conf = 65%. The above explanation was added in the revised manuscript.
6. The presentation of the paper is focusing largely on a comparison of methods. I think this should get even more prominence in the discussion.
ANSWER: Thank you for this comment. We expanded our discussion on comparison of methods. We summarized the similar findings obtained from different methods and also presented specific findings from each method. The revisions have been marked in RED.
7. I think, it would be really interesting to discuss ethnicity in more detail (e,g, that it is not included in the multiple logistic regression) because the title of the paper somehow refers to ethnicity (you mention it only briefly in the limitations paragraph) but do the results for youth in ethnic minority regions differ from other regions in China?
ANSWER: Thank you for this comment. We added a paragraph in the section of Discussion to explain results related to the role of ethnicity in this new version as follows:
Regarding the role of ethnicity, although its association with depressive symptoms was not significant using Approaches 1 and 2, ethnic minorities appeared in several association rules to predict depression risk using Approach 3. Specifically, we found that when combined with ethnic minority, the correlations with depression increased for the factor combinations of “low peer support, low family cohesion, from multiple-child family”, “low peer support and low family cohesion”, “secondary school and low family cohesion”, “low peer support, female, from multiple-child family” and “low peer support, female”, while the correlation with depression slightly decreased when ethnic minority was combined with high family conflict and low family cohesion. Generally, the results indicated that ethic minority students are generally more vulnerable to depression when exposed to multiple familial and school-related factors; however, being an ethnic minority alone did not show direct relationship with depression. This finding aligns with some studies (Zhang et al, 2018), but it contrasts with other research suggesting that ethnic minority status can be an independent risk factor for children’s development (Lee et al., 2019; Wei et al., 2022). From the perspective of the cultural-ecological framework, the effects of ethnicity depend on specific cultural ecological environments. People living in the sampling sites have a long history of multiethnic integration. Folk culture and life customs are well preserved and respected. Children learn Mandarin from an early age and have no language barriers in cross-ethnic communications. These factors might buffer the negative effects of ethnicity-related stressors to some extent. However, because the majority of participants were ethnic minority students, the class imbalance of the dataset might have led to bias in the results. In addition, because we did not measure ethnicity-related stressors directly, this issue should be explained with caution and needs further examination.
8. Last point: references 4 and 5 are identical, the same holds for 11 and 12
ANSWER: Thank you for pointing out this error. We have removed the identical references and renumbered items in the reference list.
Reviewer 2 Report
Comments and Suggestions for Authors
Dear Authors.
I find your manuscript interesting.
I have a few remarks.
Firstly, I think You should change the title to be short and clear. Avoid mentioning settings because these were not settings but rather factors you investigated.
Secondly, You should mention the age of the participants, or at least age range for every grade, since other countries have different grades at different levels of education.
There are a few grammar mistakes to be corrected.
Kind regards.
Comments on the Quality of English LanguageThere are a few grammar mistakes to be corrected.
Author Response
1. Firstly, I think You should change the title to be short and clear. Avoid mentioning settings because these were not settings but rather factors you investigated.
ANSWER: Thank you for this comment. We followed this suggestion, removed the settings and shortened the title. The revised title was “Multiple Risk Exposure in Family and School Settings and Adolescent Depressive Symptoms: Analyses Using Cumulative Risk Model, Logistic Regression and Association Rule Mining”.
2. Secondly, You should mention the age of the participants, or at least age range for every grade, since other countries have different grades at different levels of education.
ANSWER: Thank you for this comment. The mean age, standard deviation and age range has been added in the revised manuscript.
3. There are a few grammar mistakes to be corrected.
ANSWER: Thank you for this comment. We have gone through the manuscript to make sure there were no grammar mistakes.